# Genome-Driven Insights into *Lactococcus* sp. KTH0-1S Highlights Its Biotechnological Potential as a Cell Factory

**DOI:** 10.3390/biology14101323

**Published:** 2025-09-25

**Authors:** Nisit Watthanasakphuban, Hind Abibi, Nuttakan Nitayapat, Phitsanu Pinmanee, Chollachai Klaysubun, Nattarika Chaichana, Komwit Surachat, Suttipun Keawsompong

**Affiliations:** 1Department of Biotechnology, Faculty of Agro-Industry, Kasetsart University, Chatuchak, Bangkok 10900, Thailand; faginsw@ku.ac.th (N.W.); hindabibi25@gmail.com (H.A.); faginun@ku.ac.th (N.N.); 2Enzyme Technology Research Team, National Center of Genetic Engineering and Biotechnology (BIOTEC), Pathum Thani 12120, Thailand; phitsanu.pin@biotec.or.th; 3Department of Biomedical Sciences and Biomedical Engineering, Faculty of Medicine, Prince of Songkla University, Songkhla 90110, Thailand; chollachai951@gmail.com (C.K.); cnattari@medicine.psu.ac.th (N.C.)

**Keywords:** *Lactococcus* sp., genome analysis, probiotic, cell factory, safety evaluation, next-generation microbial cell factories

## Abstract

This study investigates the bacterial strain *Lactococcus* sp. KTH0-1S as a potential alternative expression host and microbial cell factory. We confirmed that it is safe, carrying no harmful genes, and possesses traits that support survival in the digestive system and fermentation environments. Uniquely, the strain naturally produces nisin, which can act as an inducer for the NICE system, providing auto-induction properties that offer a simpler and more cost-effective alternative to conventional hosts. The strain also grows efficiently with nutrient support and can utilize a variety of sugars, including those from agricultural by-products. These characteristics make *Lactococcus* sp. KTH0-1S a promising next-generation probiotic and a versatile, food-grade microbial host for producing valuable proteins.

## 1. Introduction

Lactic acid bacteria (LAB) are essential in fermented foods and recognized for their technological, nutritional, and health-promoting properties. Their historical safe consumption has earned many LAB strains “Generally Recognized as Safe” (GRAS) status from regulatory bodies like the U.S. FDA [1]. Beyond their role as starter cultures in dairy and plant-based fermentations, some LAB also function as probiotics, contributing to gut health, immune modulation, and pathogen inhibition [2,3]. This growing demand for functional foods and microbial therapeutics has spurred interest in identifying novel LAB strains with enhanced probiotic and technological traits.

Among LAB, *Lactococcus lactis* is a well-established model organism, extensively used in food biotechnology and synthetic biology. Its advantages include a small, well-annotated genome, non-pathogenic status, food-grade compatibility, and ease of genetic manipulation [4], which have facilitated the development of sophisticated expression systems. These systems make *L. lactis* a versatile microbial cell factory for producing heterologous proteins, bioactive compounds, and therapeutic molecules [5,6].

A key innovation in LAB expression technology is the Nisin-Controlled Expression (NICE) system, which uses the bacteriocin nisin as an inducer for gene expression. While the NICE system is tightly regulated and widely employed, its reliance on exogenous nisin supplementation creates challenges for industrial scalability. Adding nisin not only increases production costs but can also inhibit host cell growth due to its antimicrobial activity [7,8,9,10,11]. An appealing solution lies in using naturally nisin-producing *Lactococcus* strains. These strains carry the full nisin biosynthetic gene cluster, including immunity genes such as *nisI* and *nisFEG*, which protect the host from self-produced nisin [10]. Such strains offer the potential for auto-induction, eliminating the need for external inducers and enabling more cost-effective and scalable recombinant protein production. Furthermore, their inherent nisin immunity allows them to tolerate higher intracellular nisin concentrations, potentially enhancing expression levels without compromising cell viability. These features are particularly valuable when combined with the selection of novel *Lactococcus* strains that possess distinct genetic backgrounds, potentially offering new biosynthetic capabilities or stress-resistance traits not found in model strains.

For any LAB strain to be suitable for industrial, food, or therapeutic applications, comprehensive genomic and functional characterization is critical. This includes in silico screening for antimicrobial resistance (AMR) genes, virulence factors, and mobile genetic elements to assess biosafety [12], alongside functional analyses of probiotic traits such as stress tolerance, adhesion to intestinal cells, and biofilm formation. This integrated approach aligns with the concept of next-generation probiotics (NGPs), which are identified through genomic screening for defined, beneficial traits rather than empirical selection [13]. This allows for the discovery of novel LAB strains that combine safety, functionality, and technological potential, particularly those capable of dual roles as probiotics and hosts for recombinant protein production.

This study explores the potential of novel, auto nisin-producing *Lactococcus* strain that integrate probiotic characteristics, nisin immunity, and genetic stability. This strategy represents a promising avenue for developing next-generation microbial cell factories as alternative expression hosts suitable for both food-grade and therapeutic applications.

## 2. Material and Methods

### 2.1. Bacterial Growth, DNA Extraction, and Whole-Genome Sequencing (WGS)

*Lactococcus* sp. KTH0-1S was originally isolated from *Kung-Som*, a traditional Thai fermented shrimp product [14]. For routine cultivation, the strain was grown on M17 agar (Sigma-Aldrich, St. Louis, MO, USA) at 30 °C for 24 h. A single pure colony was subsequently transferred to M17 broth and cultivated at 30 °C for 18–24 h under static conditions (no agitation). Genomic DNA was extracted using GenElute™ Bacterial Genomic DNA kit (Sigma-Aldrich, St. Louis, MO, USA), following the manufacturer’s protocol. Briefly, 3 mL of overnight culture was harvested and resuspended in lysozyme solution, followed by incubation at 37 °C for 1 h to disrupt the peptidoglycan layer. RNase treatment was performed to remove RNA, after which proteinase K and lysis solution were added, and the sample was incubated at 55 °C for 10 min. Ethanol was then added to the lysate, which was transferred to a spin column for DNA binding. The column was washed twice with provided wash buffers, and DNA was finally eluted in nuclease-free water. DNA concentration and purity were assessed using NanoDrop spectrophotometer (ThermoFisher, Waltham, MA, USA).

Prior to whole-genome sequencing, species identification was confirmed by amplifying the 16S rRNA gene using PCR. The amplification was carried out using NEB Q5^®^ High-Fidelity 2X Master Mix (New England Biolabs, Ipswich, MA, USA) with universal primers 27F and 1492R (Table 1) as described by [15]. The amplified product (~1500 bp) was visualized via agarose gel electrophoresis and purified using the Monarch^®^ DNA Gel Extraction Kit (New England Biolabs, USA). The purified PCR product was quantified using NanoDrop spectrophotometer and submitted for Sanger sequencing. The resulting sequence was analyzed using NCBI BLAST (https://blast.ncbi.nlm.nih.gov/Blast.cgi accessed on 15 May 2024) to confirm species identity.

The confirmed genomic DNA of *Lactococcus* sp. KTH0-1S was sequenced using a hybrid long-read approach combining both PacBio Single-Molecule Real-Time (SMRT) sequencing (Newark, CA, USA) and Oxford Nanopore Technologies (ONT), (Oxford, UK). PacBio sequencing was carried out to obtain highly accurate long reads, while ONT sequencing was performed using the Rapid Barcoding Kit 24 V14 (SQK-RBK114.24) (Oxford, UK) following the manufacturer’s standard protocol. ONT sequencing was conducted in high-accuracy base-calling mode using the R10.4.1 flow cell, generating real-time long-read data. The datasets from both platforms were integrated to enhance genome assembly completeness and accuracy.

### 2.2. Genome Assembly and Annotation

The raw reads were assembled de novo using Flye version 2.9.5 [16]. The quality of the assembled genome was evaluated with QUAST version 5.3 [17]. Genome annotation was carried out using both Prokka version 1.14.6 [18] and the RAST web server [19]. A circular genome map was created for visualization and annotation using Proksee server [20]. In addition, functional annotation was performed using eggNOG-mapper v2.1.12 [21], which classified the predicted proteins into clusters of orthologous groups (COGs) [22]. Additionally, BlastKOALA v3.1 was used to assign proteins to Kyoto Encyclopedia of Genes and Genomes (KEGG) Orthology (KO) groups and to generate corresponding KEGG pathway maps [23].

### 2.3. Multilocus Sequence Typing (MLST) Phylogenetic Tree Construction

The autoMLST 2.0 (Automated Multi-Locus Species Tree) tool [24] was applied to the genome of *Lactococcus* sp. KTH0-1S, along with available *Lactococcus* type strains. The phylogenetic analysis utilized core housekeeping genes derived from the genomes using default settings with 1000 bootstrap replicates. *Streptomyces alboniger* NRRL B-1832 was used as the outgroup.

### 2.4. In Silico Safety Evaluation

The safety profile of *Lactococcus* sp. KTH0-1S was assessed through in silico analysis. Antibiotic resistance genes (ARGs) were identified using ResFinder v4.1 [25] and the Comprehensive Antibiotic Resistance Database (CARD) (https://card.mcmaster.ca accessed on 30 June 2025), implemented via ABRicate v1.0.1 with default parameters [26]. To explore potential virulence factors, the genome was queried against the Virulence Factor Database (VFDB) (https://www.mgc.ac.cn/VFs/ accessed on 7 July 2025) [27]. Homology comparisons were carried out using BLASTp with an identity cutoff of >80% against known pathogenic virulence genes. Moreover, mobile genetic elements (MGEs) and prophage regions were identified by the MobileOG database (https://mobileogdb.flsi.cloud.vt.edu accessed on 11 September 2025) and PHASTEST v3.0, respectively [28]. In addition, secondary metabolite biosynthetic gene clusters (BGCs) in the *Lactococcus* sp. KTH0-1S genome were identified using the antiSMASH 8.0 platform [24].

### 2.5. Cell Factory Potential Genes Analysis for Alternative Expression Host of Lactococcus sp. KTH0-1S

#### 2.5.1. Prediction of Bacteriocin (Nisin Z) Cluster

In this study the potential genes required for protein expression in *Lactococcus* sp. KTH0-1S, as the alternative cell factory, were identified. The nisin Z cluster genes were analyzed to check the possibility of using *Lactococcus* sp. KTH0-1S as the expression host for Nisin-Controlled Expression system (NICE), which has high efficiency for homologous and heterologous protein expression in Lactococcus. The Bacteriocin-related genes were predicted using BAGEL4 [29] and confirm the integrity of the nisin biosynthetic cluster identified in the genome sequence, PCR was carried out using primers targeting key genes (*nisZ*, *nisK*, *nisR* and *nisI*) (Table 1). These genes are particularly important for the functionality of the NICE system and the potential for auto-induction. PCR validation ensured that no sequencing or assembly artifacts occurred and provided a reliable basis for subsequent applications, such as amplification of the genes for construction of new expression plasmids tailored to *Lactococcus* sp. KTH0-1S. The amplified DNA sequences were then analyzed on 1% agarose gel electrophoresis to determine the presence of the 4 genes based on size.

#### 2.5.2. Respiration Metabolism Verification

Some *L. lactis* showed the respiration growth which resulted in much higher cell biomass formation and cell robustness [30,31], which is beneficial for cell factory applications. *Lactococcus* sp. KTH0-1S was analyzed for genes encoding all enzymes involved in the electron transport chain of *L. lactis*, including NADH dehydrogenase as electron donor, menaquinones as electron carrier, and cytochrome oxidase as the electron acceptor, using annotations based on the Kyoto Encyclopedia of Genes and Genomes (KEGG) database (http://www.genome.jp/kegg/ accessed on 20 October 2024) [32,33,34].

The respiration growth characteristic of *Lactococcus* sp. KTH0-1S was also verified. *Lactococcus* sp. KTH0-1S was cultivated in GM17 medium and incubated at 30 °C. The respiration metabolism behavior was investigated under facultative anaerobic (static) and aerobic (200 rpm) conditions, and the overnight cultures were transferred into GM17 medium with or without heme and menaquinone supplementation. The growth (OD600) in all culture conditions was collected after 24 h of incubation. The nisin activity of the cell-free supernatant from each sample was checked against a *Lactiplantibacillus plantarum* WCFS1 indicator strain, and the inhibition activity was confirmed and calculated as AU/mL using a broth microdilution assay.

#### 2.5.3. Carbohydrate-Active Enzyme Analysis

The genome of *Lactococcus* sp. KTH0-1S was analyzed for carbohydrate-active enzymes (CAZymes) using the dbCAN2 meta server, with annotations based on the CAZy database [35]. This database classifies CAZymes into six main categories: glycoside hydrolases (GHs), glycosyltransferases (GTs), carbohydrate esterases (CEs), carbohydrate-binding modules (CBMs), auxiliary activity enzymes (AAs), and polysaccharide lyases (PLs).

## 3. Results

### 3.1. Overview of the Genome in Lactococcus sp. KTH0-1S

The whole genome sequencing analysis identified the obtained strain KYH-01S as *Lactococcus* sp. The genome of this strain consists of a chromosome spanning 2,379,149 bp with a GC content of 35.1%, and a plasmid, pKTH0-1S, which is 43,284 bp with a GC content of 33.2%. A total of 2342 coding sequences (CDSs) were identified in the chromosome, while 62 CDSs were found in the plasmid. The RAST annotation system identified 236 and 3, respectively. Moreover, this strain showed the highest sequence similarity to *Lactococcus cremoris* (Accession number: GCA_004354515.1), with an average nucleotide identity (ANI) of 88.57% (Table 2). The genome visualization of *Lactococcus* sp. KTH0-1S was illustrated in Figure 1.

### 3.2. Multilocus Sequence Typing (MLST) Phylogenetic Tree

The MLST phylogenetic tree depicts the genetic relationship of *Lactococcus* sp. KTH0-1S with other *Lactococcus* strains using multilocus sequence typing based on conserved housekeeping genes. Although ANI analysis clearly indicates its taxonomic placement, MLST was additionally performed to support strain-level phylogenetic comparison with publicly available references. The resulting tree shows that *Lactococcus* sp. KTH0-1S occupies a distinct clade, genetically distant from established type strains, reinforcing its divergence within the genus (Figure 2).

### 3.3. Functional Characterization and Prediction

The functional classifications of *Lactococcus* sp. KTH0-1S were performed based on three distinct categories, including RAST subsystems, COG categories, and KO pathways. The RAST subsystems show the distribution of genomic functions, with a significant emphasis on metabolism, especially in the categories of carbohydrates, amino acids and derivatives, and protein metabolism. Other prominent categories include DNA and RNA metabolism, fatty acids, lipids, and isoprenoids, and respiration. The genetic elements category, particularly involving phages, prophages, transposable elements, and plasmids, also stands out, showing the presence of mobile genetic elements. Furthermore, smaller groups such as stress response, regulation, cell signaling, and cellular processes highlight the mechanisms for managing environmental and internal stressors of the strain (Figure 3A). However, 71% of the genome was unassigned in the RAST subsystem.

In terms of COG categories, this analysis categorizes data based on functional families, with the highest counts found in categories related to genetic information processing, such as RNA processing, translation, and transcription. Other significant categories include metabolism, particularly amino acid metabolism and energy production, and cellular processes, such as cell cycle control and intracellular trafficking. Defense mechanisms also appear, reflecting the capacity of this strain for adaptive responses to threats (Figure 3B). Moreover, KO pathways further organize the functional data, where genetic information processing and metabolism dominate the landscape. Particularly notable are pathways related to amino acid metabolism, energy metabolism, and lipid metabolism, indicating key functional roles in cellular biosynthesis and energy management. The environmental information processing and cellular processes categories also appear, while human diseases and organismal systems are represented with fewer counts, highlighting a lesser focus on these functions (Figure 3C).

### 3.4. Safety Assessment and Secondary Metabolite Biosynthetic Gene Clusters Analysis

The safety of *Lactococcus* sp. KTH0-1S was evaluated by predicting genes associated with antimicrobial resistance (AMR) and virulence factors. The strain was not found to harbor the ARG in its genome. Moreover, several virulence factor genes were identified in *Lactococcus* sp. KTH0-1S. The *cpsI* gene, which is involved in immune modulation, can help the strain modulate the immune system of host, potentially providing protective effects against pathogens. The *tufA* gene, encoding elongation factor Tu, plays a role in adhesion, which is important for the bacteria to effectively colonize the gut and outcompete harmful microbes. Similarly, *htpB*, a heat shock protein, aids in adherence, improving the stability and persistence of the strain in the gastrointestinal tract. Furthermore, genes like *cps4I*, which are involved in the synthesis of capsular polysaccharides, contribute to immune modulation and protect the strain from host defenses. The *fbp54* gene, encoding fibronectin-binding protein, also supports adherence to intestinal cells. Genes, such as *hasC*, involved in the synthesis of *N*-acetylglucosamine, promote immune modulation, potentially benefiting gut health by supporting the natural immune response of the host (Table 3).

The antiSMASH analysis revealed multiple biosynthesis gene clusters for secondary metabolite production across seven regions, which are crucial for its antimicrobial properties. Terpene precursor biosynthesis clusters were identified in regions 1, 5, and 6, while region 2 encodes the RiPP Recognition Element (RRE). Region 3 is associated with betalactone synthesis, region 4 with lanthipeptide-class-I production, and region 7 with type III polyketide synthase (T3PK) biosynthesis, as shown in Figure 4.

### 3.5. Identification of Mobile Genetic Element (MGE) and Prophage Region

The MGE analysis of *Lactococcus* sp. KTH0-1S identified 88 genes classified into distinct functional categories. The majority of genes were associated with replication, recombination, and repair (29 genes), followed by integration and excision (22 genes), including integrases and recombinases. Additional categories included phage and transfer-related genes, which found 19 and 12 genes, respectively. A smaller number of genes were linked to stability, transfer, and defense mechanisms (6 genes) (Figure 5). Among these, integration-related genes included integrases (*int2*, *int-Tn*), a transposase (*tnpA*), and a site-specific recombinase (*xerS*), which are involved in the movement and integration of genetic elements. Phage-associated genes such as *clpB*, *clpP*, *clpX*, *dnaK*, *cro*, *kilA*, *xhlB*, *orf45*, *tmk*, and *oppA* suggest the presence of prophage elements. Replication and repair functions were represented by core genes including *dnaB*, *dnaH*, *dnaJ*, *ftsZ*, *gyrA*, *gyrB*, *mutS*, *radA*, *rarA*, *recA*, *recR*, *recU*, *rnhB*, *rnj*, *ruvB*, *ssb*, *topA*, *umuC*, *uvrA*, *uvrB*, *xseA*, and *xth*, which are essential for genome maintenance. Stability and defense-related genes such as *ardA*, *dcm*, *hsdM*, *hsdR*, *rex*, and *ycbY* were also detected, along with transfer-related genes including *copR*, *dut*, *groL*, *oppB*, *oppC*, *oppD*, and *oppF*, which are involved in metal resistance, DNA metabolism, and transport systems (Appendix A). However, several genes were unclassified (unknown) in the MGE category, indicating mobile element-related sequences that could not be confidently assigned to a specific functional group based on current databases. Moreover, two prophage regions were identified in the genome of *Lactococcus* sp. KTH0-1S, both classified under the Siphoviridae family and marked as non-transposable, representing various phage functions, including structural, replication, lysis, and regulatory proteins (Appendix A).

### 3.6. Identification of Probiotic Marker Genes

The identification of probiotic genes in the *Lactococcus* sp. KTH0-1S genome revealed several key functions associated with stress tolerance, adhesion, biofilm formation, and nutrient acquisition. Genes such as *dnaK*, *clpP*, *groL*, and *dnaJ* are involved in stress response mechanisms, ensuring survival under adverse conditions of the strain. The *gapA*, *pgi*, and *tpiA* genes support energy metabolism and adhesion, both essential for colonization in the gut. While *luxS* and *glf2* genes play roles in biofilm formation, with *luxS* contributing to quorum sensing and *glf2* to exopolysaccharide production. The *eno* aids in metabolism, supporting biofilm development. Moreover, *lepA*, *bglH*, *pepT*, *tuf*, *padC*, *gapA*, and *fusA* are involved in nutrient acquisition, enabling the strain to utilize various substrates, enhancing its ability to thrive in diverse environments (Table 4).

### 3.7. Exopolysaccharide (EPS)-Associated Gene Identification

The *Lactococcus* sp. KTH-01S genome contains several genes associated with EPS biosynthesis and modification. *epsA* functions as a transcriptional activator for EPS biosynthesis. *epsB* is a manganese-dependent protein–tyrosine phosphatase, while *epsC* and epsD encode tyrosine-protein kinases involved in modulating EPS production. *epsE* is a galactosephosphotransferase responsible for transferring galactose to undecaprenyl-phosphate, a key step in EPS synthesis. *epsF* and *epsH* are involved in glycosyltransferase and acetyltransferase activities, respectively. *lytR* acts as an antiterminator, regulating the transcription of EPS genes. Genes related to capsular polysaccharide synthesis, including *cpsA*, *cpsB*, *cpsC*, *cpsD*, *cpsH*, and *cpsE*, are involved in sugar transfer, polysaccharide export, and exopolysaccharide synthesis. The *glt2* and *glt1* encode glycosyltransferases, while *licD3* is involved in lipopolysaccharide modification (Table 5).

### 3.8. Identification of Bacteriocin Biosynthetic Clusters

Comprehensive genome analysis of *Lactococcus* sp. KTH0-1S using the BAGEL4 tool identified multiple gene clusters associated with bacteriocin biosynthesis, including those responsible for the production of lantibiotic-class antimicrobial peptides (Figure 6). Notably, a nisin Z biosynthetic gene cluster was detected (green), encompassing key genes involved in precursor modification, such as *lanB* and *lanC* (blue), which mediate dehydration and cyclization reactions essential for nisin maturation.

Regulatory elements (yellow) within the cluster include the two-component system comprising the response regulator (*lanR*) and the histidine kinase sensor (*lanK*), which coordinately regulate transcription of the nisin operon in response to environmental stimuli. A gene encoding a nisin leader peptide-processing serine protease (purple) was also identified, facilitating the cleavage of the leader sequence to yield the mature, bioactive peptide.

In addition, genes involved in immunity and export mechanisms were annotated. These include the ABC-type nisin transport ATP-binding protein, the nisin immunity protein (lipopeptide; *orf00023*), and an ABC transporter-like system (*orf00031*, *orf00033*), marked in red (Figure 6). Collectively, these findings highlight the genetic potential of *Lactococcus* sp. KTH0-1S for the production, regulation, and self-protection against the lantibiotic nisin Z.

PCR amplification was performed to confirm the presence of key nisin-associated genes (*nisZ*, *lanK*, *lanR*, and *lanI*) in the genome of *Lactococcus* sp. KTH0-1S. All four genes were successfully amplified, verifying their presence in the chromosomal DNA (Appendix A). The confirmed presence of these genes supports the strain’s potential for nisin production and its suitability as a candidate for use in biotechnological applications.

### 3.9. Respiration Metabolism Verification

Functional annotation of the *Lactococcus* sp. KTH0-1S genome using the Kyoto Encyclopedia of Genes and Genomes (KEGG) database revealed the presence of genes encoding the core components required for respiration metabolism in *L. lactis*. Specifically, the strain harbors the *ndh* gene encoding NADH dehydrogenase, which initiates the electron transport process by transferring electrons from NADH to quinones (Table 6). In addition, a complete set of genes involved in the biosynthesis of menaquinone (vitamin K2), including *menA*, *menB*, *menC*, *menD*, *menE*, *menF*, and *menH* were identified. Menaquinone functions as a key electron carrier in the electron transport chain (ETC) of Gram-positive bacteria, including lactic acid bacteria [30], linking dehydrogenase activity to terminal oxidases [31].

Moreover, genes encoding the cytochrome bd-type oxidase complex (*cydA*, *cydB*, *cydC*, *cydD*) were also detected (Table 6). This oxidase complex serves as the terminal electron acceptor in the ETC, facilitating oxygen reduction under low-oxygen (microaerobic) conditions. The presence of these genes strongly suggests that *Lactococcus* sp. KTH0-1S possesses the genetic capacity for respiration metabolism, provided that essential cofactors such as heme and menaquinone are available similar to the respiration-enabled *L. lactis* strains described previously [30,31].

### 3.10. Growth Behavior and Respiratory Activation

The growth of *Lactococcus* sp. KTH0-1S was evaluated under different cultivation conditions to assess its respiratory potential. Under anaerobic conditions, the strain exhibited moderate growth with a final OD600 of 1.257 ± 0.07 and a decrease in pH to 5.02 ± 0.03 (Figure 7). Aerobic conditions alone increased the final OD600 to 1.823 ± 0.02. Supplementation with heme under aerobic conditions resulted in the highest biomass accumulation (OD600 = 2.413 ± 0.07), while the addition of menaquinone (MK-4) along with heme did not lead to a significant further increase (OD600 = 2.408 ± 0.07), indicating that heme was the primary limiting factor for respiratory activation on this strain. Similar trends were observed in the final culture pH, with the highest values recorded in aerobic cultures supplemented with heme or heme + MK-4 were pH = 5.52 ± 0.01 and 5.54 ± 0.2, indicating reduced acidification.

Nisin activity from *Lactococcus* sp. KTH0-1S was measured in various cultured conditions. The highest antimicrobial activity was observed in anaerobic + heme (14.00 ± 0.50 mm) and anaerobic (13.50 ± 0.87 mm) conditions (Table 7). Slightly reduced inhibition zones were observed under respiratory conditions, particularly in aerobic + heme + MK-4 (11.00 ± 0.50 mm), but the strain remained active in nisin production across all tested growth modes.

### 3.11. Carbohydrate-Active Enzyme (CAZyme) Prediction

CAZyme families were then identified in the *Lactococcus* sp. KTH-01S genome by mapping to the CAZy database using the dbCAN2 server. The results showed that the glycosyltransferase (GT) family was the most abundant, with families GT2, GT1, GT4, GT51, GT5, GT35, and GT28. Several glycoside hydrolase (GH) families were also present, including GH13, GH31, GH65, GH8, GH43, GH38, GH32, GH20, and GH18. The genome also contained carbohydrate esterase (CE) family CE1, carbohydrate-binding modules (CBM) families CBM50, CBM48, and CBM73, and auxiliary activities (AA) family AA10 (Figure 8).

## 4. Discussion

The MLST analysis of *Lactococcus* sp. KTH0-1S revealed that it is genetically distinct from other reference strains within the *Lactococcus* genus. The phylogenetic tree showed that *Lactococcus* sp. KTH0-1S occupies a unique position, which suggests that it represents a novel strain with distinct genetic characteristics compared to other type strains. This genetic uniqueness highlights its potential for novel applications in the probiotic and biotechnological fields.

The genome of *Lactococcus* sp. KTH0-1S was extensively analyzed to assess its potential as a probiotic, including its antimicrobial properties, safety profile, and ability to perform essential probiotic functions such as adhesion, biofilm formation, and stress tolerance. The functional annotation of *Lactococcus* sp. KTH0-1S revealed a genomic landscape rich in genes related to core metabolic and cellular processes, suggesting the robust metabolic flexibility and adaptive capabilities of this strain. RAST subsystem classification emphasized a predominant role of metabolism, especially in carbohydrate, amino acid, and protein metabolism, which are essential for energy production, biosynthesis, and survival under varying environmental conditions. The presence of genes involved in DNA and RNA metabolism, fatty acid and lipid biosynthesis, and respiration further underscores the strain’s potential for growth and maintenance in diverse niches. Notably, a substantial proportion of the genome (71%) remained unassigned to known subsystems, implying the existence of many hypothetical or uncharacterized genes that may contribute to unique strain-specific features or probiotic traits not yet captured in reference databases. Compared to the closest strain, *L. cremoris* (GCA_004354515.1), which exhibited 75% of unassigned CDSs in the RAST subsystem, *Lactococcus* sp. KTH0-1S showed a similar trend, indicating that a large portion of the genome in both strains may consist of hypothetical or uncharacterized genes, reflecting limited functional annotation and potential strain-specific features. In addition, COG and KO pathway analyses further confirmed dominant roles in genetic information processing, energy metabolism, and cellular processes, supporting the strain’s potential resilience, adaptability, and suitability as a probiotic.

In silico safety analysis was performed to evaluate the presence of antimicrobial resistance (AMR) genes and virulence factors. The results showed that *Lactococcus* sp. KTH0-1S does not harbor any significant AMR genes, indicating that it poses a minimal risk of transferring resistance to harmful pathogens [36,37,38]. In addition, the strain was found to possess a low number of virulence-associated genes, with no significant genes linked to pathogenicity.

Genes identified in *Lactococcus* sp. KTH0-1S genome, are increasingly recognized in the context of probiotic function due to their roles in host-microbe interactions. The *cpsI* and *cps4I* genes, involved in capsular polysaccharide biosynthesis, may contribute to immune modulation and provide protection against environmental stressors within the gastrointestinal tract, enhancing bacterial persistence [39]. Moreover, the *tufA* gene, encoding elongation factor Tu, and *htpB*, a heat shock protein, have been reported to function as surface-associated adhesion factors in non-pathogenic lactic acid bacteria, facilitating colonization of the intestinal mucosa [40]. Similarly, *fbp54*, encoding a fibronectin-binding protein, may support adhesion to epithelial cells, a key feature for probiotic efficacy [41], while the *hasC* gene, implicated in the synthesis of N-acetylglucosamine, may also play a role in modulating host immune responses and maintaining mucosal integrity [42]. These genes are annotated and more consistent with attributes beneficial for probiotic activity, including adhesion, immune interaction, and environmental resilience, rather than pathogenicity. This suggests that *Lactococcus* sp. KTH0-1S is safe for consumption and does not have the genetic makeup that would pose a risk to human health. The identification of 88 MGE-associated genes in *Lactococcus* sp. KTH0-1S highlights the dynamic nature of its genome and reflects its evolutionary potential. The high proportion of genes related to replication, recombination, and repair suggests that the strain maintains a robust system for preserving genome integrity and accommodating genomic rearrangements. The presence of integrases, transposases, and recombinases within the integration/excision category indicates the potential for site-specific recombination and the mobility of genetic elements, which may facilitate adaptation to changing environments [43]. Furthermore, the detection of numerous phage-related genes, along with two integrated prophage regions, supports the notion that phage interactions have played a role in shaping the genome structure. The prophage elements, particularly those classified under the Siphoviridae family, carry a variety of functional components, including structural proteins, lysis enzymes, and transcriptional regulators, reflecting their potential involvement in lysogenic conversion [44]. The presence of defense- and restriction–modification-related genes, such as *hsdR*, *hsdM*, *ardA*, and *dcm*, suggests that the strain possesses systems for protecting against foreign DNA, while transfer-associated genes such as copR and oligopeptide transporter components (oppB-F) may contribute to environmental sensing and nutrient uptake [45]. The detection of several unclassified MGE-associated genes further indicates the presence of potentially novel or poorly characterized mobile elements, underscoring the need for continued database expansion and functional validation. The MGE profile of *Lactococcus* sp. KTH0-1S reflects a flexible and adaptive genome structure that may support its ecological fitness and functional roles in complex environments.

A comprehensive analysis of the *Lactococcus* sp. KTH0-1S genome revealed several genes linked to key probiotic properties, including stress tolerance, adhesion, biofilm formation, and nutrient acquisition. Stress tolerance is facilitated by genes like *dnaK*, *clpP*, *groL*, and *dnaJ* [46], which help the strain withstand environmental stresses such as heat, oxidative stress, and low pH [47]. These genes encode chaperone proteins and proteases that assist in maintaining protein stability and promoting protein folding, which are essential for bacterial survival under the harsh conditions encountered in the gastrointestinal tract, particularly in the stomach and small intestine [48]. For adhesion to intestinal cells, genes like *gapA*, *pgi*, and *tpiA* [49], involved in glycolysis and energy production, indirectly support adhesion by providing the metabolic energy needed for this process [49,50]. Adhesion is crucial for probiotics to establish a stable colony in the gut, and these genes contribute to the strain’s ability to bind to intestinal epithelial cells, ensuring that it can adhere to the gut lining and resist being washed out during intestinal transit [51]. Biofilm formation, an important feature for probiotics, is supported by the *luxS* gene, which is involved in quorum sensing, enabling bacteria to communicate and regulate gene expression in response to population density [52]. This mechanism is essential for biofilm formation, which offers protection from environmental stresses like bile salts and antimicrobial agents [53]. The *glf2* gene encodes UDP-galactopyranose mutase, which is involved in the synthesis of polysaccharides for biofilm formation [54], suggesting that *Lactococcus* sp. KTH0-1S can form biofilms, enhancing its stability and survival in the gastrointestinal tract. Regarding nutrient acquisition and metabolism, genes such as *lepA*, *bglH*, *pepT*, *tuf*, *padC*, *gapA*, and *fusA* contribute to various aspects of nutrient acquisition, transport, and metabolism. The *lepA* encodes a divalent copper transporter, essential for maintaining metal ion homeostasis, which is crucial for enzymatic activities and overall cellular function [55]. The *bglH* is involved in utilizing beta-glucosides, which are abundant in plant-based foods, allowing the strain to thrive in diverse dietary environments [56]. *pepT* aids in the breakdown of peptides, facilitating nutrient acquisition [56,57], while *padC* is involved in phenolic acid metabolism, relevant for the strain’s survival on plant-based substrates in the gut [58]. The ability to metabolize phenolic compounds could further enhance the strain’s fitness in the gastrointestinal tract, where such compounds are commonly found in the diet.

The identification of a complete nisin Z biosynthesis gene cluster in *Lactococcus* sp. KTH0-1S supports its potential not only as a natural producer of antimicrobial peptides but also as a promising alternative host for the Nisin-Controlled Expression (NICE) system, which is widely used in *L. lactis* for regulated expression of heterologous proteins, relies on a two-component quorum-sensing mechanism involving the *nisR* and *nisK* regulatory genes and external supplementation of nisin as an inducer [10]. In *Lactococcus* sp. KTH0-1S, the native presence of *nisZ* (structural gene), *lanB* and *lanC* (post-translational modification enzymes), *nisP* (leader peptidase), as well as the regulatory (*lanK*, *lanR*) and immunity genes (*orf00023*, *orf00031*, *orf00033*) suggests that this strain can autonomously produce, regulate, and tolerate nisin expression [10]. This auto-induction capability offers significant advantages for biotechnological applications. Unlike the model NICE host *L. lactis* NZ9000, which requires precise external dosing of purified nisin to initiate gene expression, *Lactococcus* sp. KTH0-1S can likely initiate expression endogenously as cell density increases and nisin accumulates in the culture medium. This reduces the need for expensive nisin supplementation, simplifies process development, and enables more scalable and reproducible expression workflows. In cost-sensitive industries such as food, feed, and nutraceutical production, auto-induction significantly enhances process economics.

Furthermore, the presence of complete nisin immunity mechanisms in *Lactococcus* sp. KTH0-1S strengthens its candidacy for recombinant expression. In traditional NICE hosts, high-level nisin exposure can lead to growth inhibition or cell lysis due to the lack of strong resistance elements. In contrast, KTH0-1S encodes the lipoprotein *nisI (orf00023)* and the ATP-binding cassette (ABC) transporter complex *nisFEG* (*ABC*, *orf00031*, *orf00033*), which together protect the host from the pore-forming activity of nisin by sequestering it and actively exporting it [59]. This allows KTH0-1S to tolerate higher nisin concentrations, facilitating stronger induction signals and higher target gene expression without compromising cell viability. This feature is especially critical when overexpressing toxic proteins or aiming for high volumetric productivity in industrial fermenters. Another important advantage of KTH0-1S is the chromosomal localization of the nisin biosynthetic operon. Unlike plasmid-based systems, which require antibiotic selection and are subject to genetic instability during prolonged or large-scale cultivation, chromosomally integrated expression cassettes offer greater genetic stability. This makes the strain more suitable for food-grade applications, where the use of antibiotic resistance markers and plasmid maintenance systems is restricted or undesirable.

In addition to its suitability as a production chassis, genome analysis of KTH0-1S revealed a diverse set of carbohydrate-active enzymes (CAZymes), including glycoside hydrolases (GHs), glycosyltransferases (GTs), and carbohydrate esterases (CEs). These enzymes confer the ability to degrade and utilize a wide variety of complex carbohydrates, positioning the strain for flexible growth on diverse carbon sources, including inexpensive agricultural or food industry by-products. For instance, GH families such as α-amylases and glucoamylases enable the hydrolysis of starchy wastes like potato peels and cassava pulp into fermentable sugars [60]. Similarly, the presence of cellulases, xylanases, and acetylxylan esterases allows for partial utilization of lignocellulosic substrates such as wheat straw, rice husks, or sugarcane bagasse provided appropriate pre-treatment is performed to reduce recalcitrance [61].

The combination of metabolic flexibility and robust expression capabilities makes KTH0-1S highly suitable for low-cost, high-density fermentations, which are critical for commercial biomanufacturing. Its ability to grow under both fermentative and respiratory conditions (as shown in earlier results) further enhances its robustness, stress tolerance, and biomass yield, particularly when supplemented with heme. This opens opportunities for aerobic or semi-aerobic fed-batch strategies, which are often preferred in industrial fermenters for improved process control and oxygen availability.

## 5. Conclusions

*Lactococcus* sp. KTH0-1S is a promising alternative to conventional *L. lactis* expression hosts. Its naturally integrated nisin gene cluster provides autonomous induction and immunity; its genomic features support broad carbohydrate metabolism; and its stable chromosomal traits reduce operational complexity. Together, these attributes support the development of *Lactococcus* sp. KTH0-1S as a next-generation, food-grade cell factory and efficient expression host for sustainable and scalable bioproduction.

## Figures and Tables

**Figure 1 biology-14-01323-f001:**
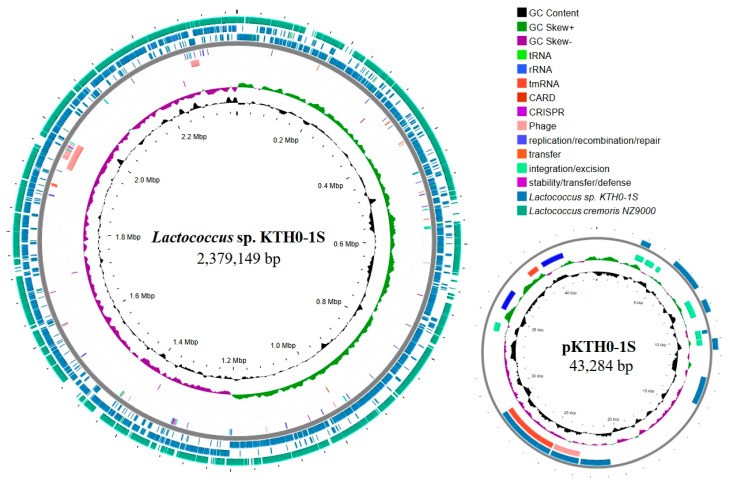
Circular genome maps of *Lactococcus* sp. KTH0-1S and its plasmid pKTH0-1S. The outermost circle represents the comparison with *Lactococcus cremoris* NZ9000, *Lactococcus* sp. KTH0-1S genome, mobile genetic elements (MGEs), phage, antimicrobial resistance genes (ARGs), CRISPR-Cas, GC skew (positive and negative), and GC content.

**Figure 2 biology-14-01323-f002:**
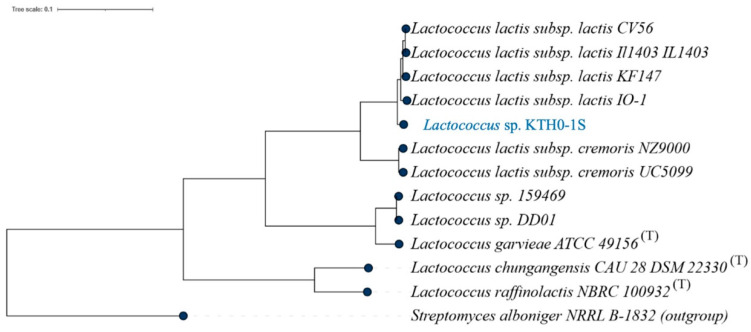
MLST phylogenetic tree showing the genetic relationship of *Lactococcus* sp. KTH0-1S with other strains based on multilocus sequence typing analysis.

**Figure 3 biology-14-01323-f003:**
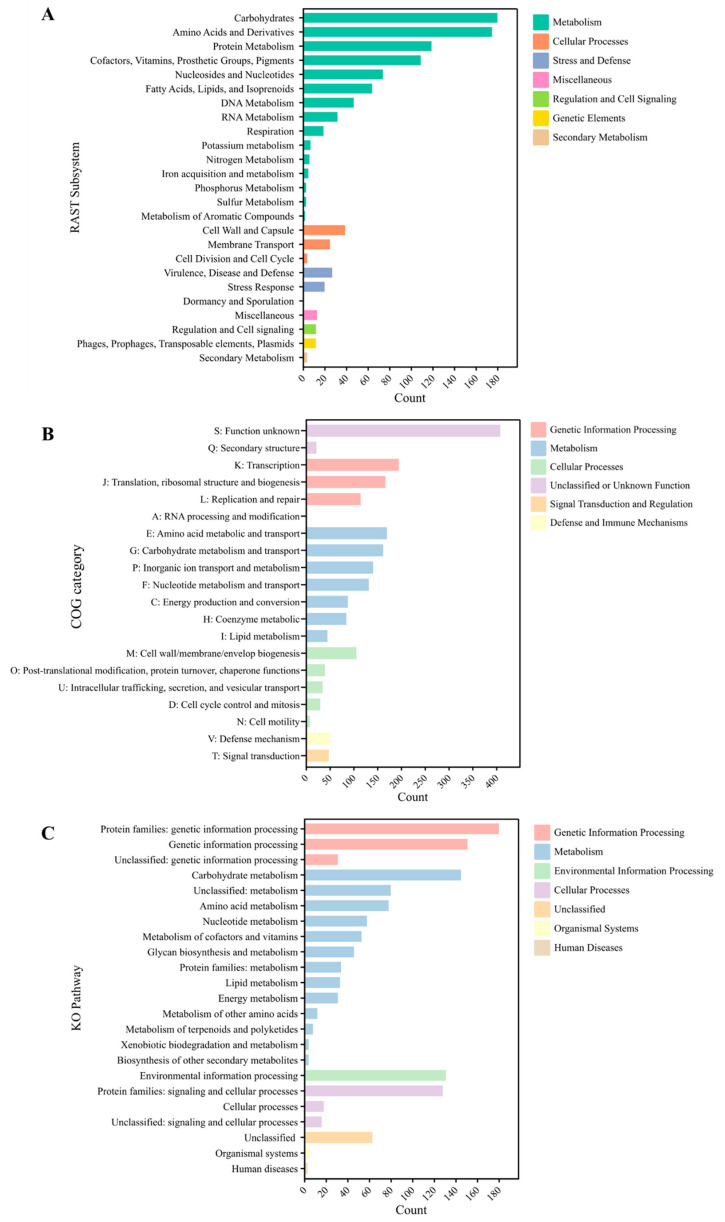
Functional classification of KTH0-1S genomic features. (**A**) RAST subsystems, (**B**) COG categories, and (**C**) KO pathways.

**Figure 4 biology-14-01323-f004:**
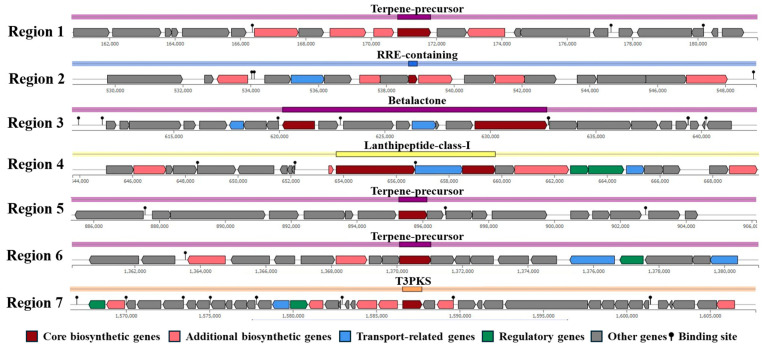
Identification of secondary metabolites biosynthesis gene clusters in *Lactococcus* sp. KTH0-1S genome.

**Figure 5 biology-14-01323-f005:**
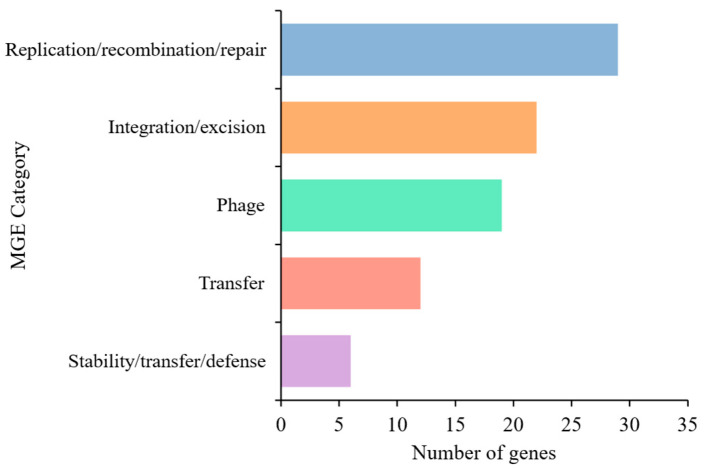
Distribution of MGE categories identified in the *Lactococcus* sp. KTH0-1S genome using the mobileOG-db tool.

**Figure 6 biology-14-01323-f006:**
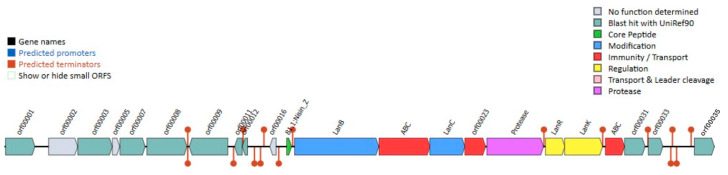
Identification of bacteriocin biosynthesis gene clusters in *Lactococcus* sp. KTH0-1S genome.

**Figure 7 biology-14-01323-f007:**
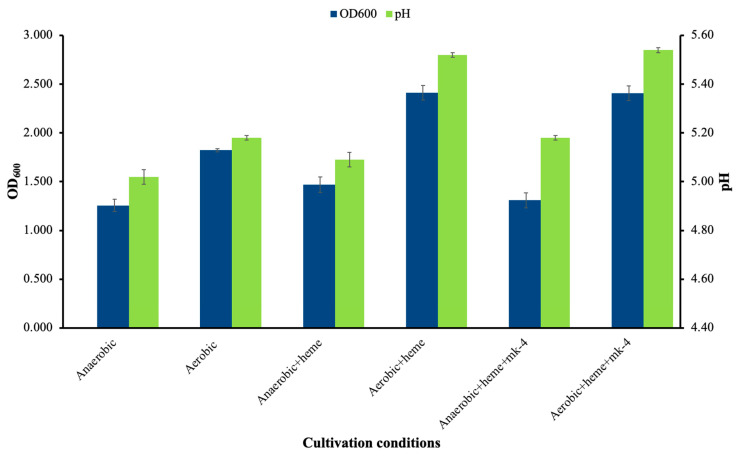
Cell growth and pH of *Lactococcus* sp. KTH-01S under different growth conditions.

**Figure 8 biology-14-01323-f008:**
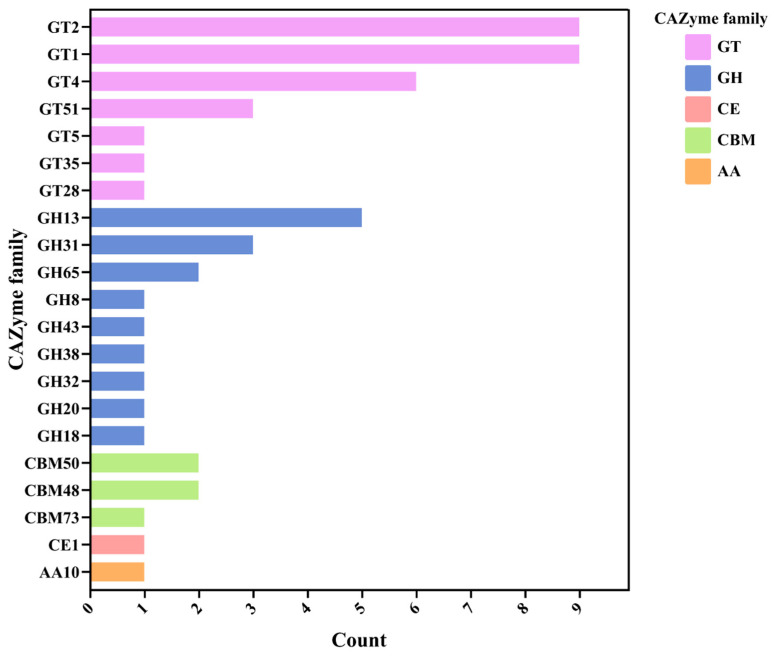
Identification of CAZyme families in the *Lactococcus* sp. KTH-01S genome. The bar chart shows the distribution of various CAZyme families, including GT (glycosyltransferase), GH (glycoside hydrolase), CE (carbohydrate esterase), CBM (carbohydrate-binding modules), and AA (auxiliary activities).

**Table 1 biology-14-01323-t001:** Primers designed and used in this study.

Primers	Sequence	References
Primers used for strain identification		
27F	AGAGTTTGATCCTGGCTCAG	[15]
1492R	GGTTACCTTGTTACGACTT	[15]
Primers for PCR verification of nisin genes in *Lactococcus* sp. KTH0-1S		
NisZ F	ATGAGTACAAAAGATTTTAACTTGG	This study
NisZ R	TTATTTGCTTACGTGAATACTACA	This study
NisK F	ATGGGTAAAAAATATTCAATGCGT	This study
NisK R	CTACACCTTGAGCAAAAGATAGT	This study
NisR F	ATGAAGACAGCATTAGAAATGAGA	This study
NisR R	TTACCCATGCCACTGATACC	This study
NisI F	ATGAGAAAATATTTAATACTTATTGTGGC	This study
NisI R	CAGCTAAAAAAATCCTGATATTCTCC	This study

**Table 2 biology-14-01323-t002:** Genome features and information of *Lactococcus* sp. KTH0-1S.

Genome Features	Chromosome	Plasmid pKTH0-1S
Genome size (bp)	2,379,149	43,284
GC content (%)	35.1	33.2
Number of contigs	1	1
L50	1	1
Number of CDSs	2342	62
tRNA	64	0
rRNA	19	0
tmRNA	1	0
Number of subsystems	236	3
Probability of being a human pathogen	0.211	
Classification	*Lactococcus* sp.	
Closest placement reference	*Lactococcus cremoris *(GCA_004354515.1)	
Average nucleotide identity (ANI)	88.57%	

**Table 3 biology-14-01323-t003:** Identification of virulence factor-associated genes found in the *Lactococcus* sp. KTH0-1S genome.

Gene	Description	Function	Locus Tag	Accession ID	Identity	E-Value
*cpsI*	UDP-galactopyranose mutase	Immune modulation	FLKGDHHP_00215	WP_002376666	79%	9 × 10^−19^
*tufA*	Elongation factor Tu	Adherence	FLKGDHHP_01848	WP_003028672	84%	8 × 10^−16^
*clpE*	ATP-dependent protease	Stress survival	FLKGDHHP_00516	NP_464522	80%	3 × 10^−12^
GBS_RS06585	UDP-N-acetylglucosamine-LPS N-acetylglucosamine transferase	Immune modulation	FLKGDHHP_01416	WP_000686634	81%	8 × 10^−10^
*hasC*	UTP-glucose-1-phosphate uridylyltransferase HasC	Immune modulation	FLKGDHHP_01336	WP_010922799	82%	7× 10^−7^
*clpC*	Endopeptidase Clp ATP-binding chain C	Stress survival	FLKGDHHP_00642	NP_463763	84%	3 × 10^−6^
*htpB*	Hsp60, 60K heat shock protein HtpB	Adherence	FLKGDHHP_01488	WP_197535493	82%	1 × 10^−5^
*cps4I*	Capsular polysaccharide biosynthesis protein Cps4I	Immune modulation	FLKGDHHP_01203	WP_000758382	80%	5 × 10^−5^
*clpB/vasG*	Type VI secretion system AAA+ family ATPase	Effector delivery system	FLKGDHHP_01494	WP_000619136	89%	0.003
*fbp54*	Fibronectin-bing protein Fbp54	Adherence	FLKGDHHP_00255	WP_002991968	92%	0.003

**Table 4 biology-14-01323-t004:** Key genes associated with probiotic properties identified in *Lactococcus* sp. KTH0-1S genome.

Gene	Start (bp)	End (bp)	Function	Locus Tag	Identity	Accession ID
Stress tolerance						
*dnaK*	984,832	986,440	Chaperone protein DnaK	FLKGDHHP_00983	72.1%	WP_031538331.1
*clpP*	638,007	638,595	ATP-dependent Clp protease proteolytic subunit	FLKGDHHP_01900	70.0%	WP_005684526.1
*groL*	390,355	391,917	ATP-dependent protein folding chaperone	FLKGDHHP_00395	69.4%	WP_007496015.1
*dnaJ*	2,320,721	2,321,782	Heat shock protein binding	FLKGDHHP_02304	67.5%	WP_002289374.1
Adhesion						
*gapA*	498,896	499,879	Glyceraldehyde-3-phosphate dehydrogenase	FLKGDHHP_02326	67.2%	WP_002901243.1
*pgi*	2,247,026	2,248,214	Glucose-6-phosphate isomerase	FLKGDHHP_02214	68.6%	WP_031538331.1
*tpiA*	1,114,463	1,115,182	Triosephosphate isomerase	FLKGDHHP_01107	71.6%	WP_002964266.1
*eno*	616,230	617,481	Phosphopyruvate hydratase activity	FLKGDHHP_00276	69.5%	WP_002964261.1
Biofilm formation						
*luxS*	258,331	258,645	S-ribosylhomocysteine lyase	FLKGDHHP_00265	70.9%	WP_005685498.1
*glf2*	210,874	211,601	UDP-galactopyranose mutase		67.4%	WP_021036608.1
Nutrient acquisition and utilization						
*lepA*	1,084,998	1,086,755	P-type divalent copper transporter activity	FLKGDHHP_00215	68.9%	Q2FXY7
*bglH*	1,701,729	1,702,073	Aryl-phospho-beta-D-glucosidase BglH	FLKGDHHP_00700	70.6%	WP_004866872.1
*pepT*	1,865,899	1,866,907	Peptidase T	FLKGDHHP_01797	66.2%	WP_004150810.1
*tuf*	1,917,440	1,918,617	Elongation factor Tu	FLKGDHHP_01848	76.3%	Q5XD49
*padC*	2,038,021	2,038,404	Phenolic acid decarboxylase Pad	FLKGDHHP_01993	77.7%	O07006
*typA*	2,111,854	2,113,671	Large ribosomal subunit assembly factor BipA	FLKGDHHP_02076	70.9%	P32132
*fusA*	2,365,432	2,367,527	Glucan 1,4-alpha-malthydrolase activity	FLKGDHHP_02350	72.0%	WP_002920103.1
Oxidative stress						
*uvrA*	1,874,208	1,877,010	Zinc ion binding and ATP hydrolysis activity	FLKGDHHP_01805	67.3%	WP_002296519.1

**Table 5 biology-14-01323-t005:** EPS biosynthesis genes found in *Lactococcus* sp. KTH-01S genome.

Gene	Functional Role
*epsA*	Exopolysaccharide biosynthesis transcriptional activator EpsA
*epsB*	Manganese-dependent protein-tyrosine phosphatase
*epsC*	Tyrosine–protein kinase transmembrane modulator EpsC
*epsD*	Tyrosine–protein kinase EpsD
*epsE*	Undecaprenyl–phosphate galactosephosphotransferase
*epsF*	Exopolysaccharide biosynthesis glycosyltransferase EpsF
*epsH*	Exopolysaccharide biosynthesis acetyltransferase EpsH
*lytR*	Exopolysaccharide biosynthesis transcription antiterminator, LytR family
*cpsA*	Capsular polysaccharide synthesis enzyme CpsA, sugar transferase
*cpsB*	Capsular polysaccharide synthesis enzyme CpsB
*cpsC*	Capsular polysaccharide synthesis enzyme CpsC, polysaccharide export
*cpsD*	Capsular polysaccharide synthesis enzyme CpsD, exopolysaccharide synthesis
*cpsH*	Capsular polysaccharide synthesis enzyme CpsH
*glt2*	Glycosyl transferase, group 2 family protein
*glt1*	Glycosyl transferase, group 1 family protein
*UP*	Putative uncharacterized protein in cluster with two glycosyl transferases
*CP*	Conserved domain protein in cluster with two glycosyl transferases
*licD3*	Lipopolysaccharide cholinephosphotransferase LicD3
*cpsE*	Capsular polysaccharide synthesis enzyme CpsE

**Table 6 biology-14-01323-t006:** Relevant genes related with respiration metabolism identified in *Lactococcus* sp. KTH0-1S genome.

Gene	KEGG Gene ID	Description	Function	Locus Tag	Identity	E-Value
**Electron donor**						
*ndh*	lla:L39857	NADH dehydrogenase	Metabolism; Energy metabolism; Oxidative phosphorylation	FLKGDHHP_00343	99.8%	1.70 × 10^−237^
*ndh*	lls:lilo_0760	NADH dehydrogenase	Metabolism; Energy metabolism; Oxidative phosphorylation	FLKGDHHP_00851	99.9%	0.00 × 10^0^
**Electron carrier**						
*menC*	llj:LG36_0709	O-succinylbenzoate synthase	Metabolism; Metabolism of cofactors and vitamins; Ubiquinone and other terpenoid-quinone biosynthesis	FLKGDHHP_00756	99.5%	1.10 × 10^−208^
*menE*	lls:lilo_0660	O-succinylbenzoic acid---CoA ligase	Metabolism; Metabolism of cofactors and vitamins; Ubiquinone and other terpenoid-quinone biosynthesis	FLKGDHHP_00648	99.3%	3.30 × 10^−255^
*menB*	lls:lilo_0661	naphthoate synthase	Metabolism; Metabolism of cofactors and vitamins; Ubiquinone and other terpenoid-quinone biosynthesis	FLKGDHHP_00758	100%	1.60 × 10^−159^
*menH*	llx:NCDO2118_0746	2-succinyl-6-hydroxy-2,4-cyclohexadiene-1-carboxylate synthase	Metabolism; Metabolism of cofactors and vitamins; Ubiquinone and other terpenoid-quinone biosynthesis	FLKGDHHP_00759	98.5%	6.50 × 10^−150^
*menD*	llj:LG36_0713	2-succinyl-5-enolpyruvyl-6-hydroxy-3-cyclohexene-1-carboxylate synthase	Metabolism; Metabolism of cofactors and vitamins; Ubiquinone and other terpenoid-quinone biosynthesis	FLKGDHHP_00760	99.1%	0.00 × 10^0^
*menF*	llj:LG36_0714	menaquinone-specific isochorismate synthase	Metabolism; Metabolism of cofactors and vitamins; Ubiquinone and other terpenoid-quinone biosynthesis, Metabolism; Metabolism of terpenoids and polyketides; Biosynthesis of siderophore group nonribosomal peptides	FLKGDHHP_00761	99.7%	6.40 × 10^−220^
*ubiE (menG)*	llj:LG36_1619	demethylmenaquinone methyltransferase/2-methoxy-6-polyprenyl-1,4-benzoquinol methylase	Metabolism; Metabolism of cofactors and vitamins; Ubiquinone and other terpenoid-quinone biosynthesis	FLKGDHHP_01643	99.6%	4.70 × 10^−139^
*menA*	lld:P620_01240	1,4-dihydroxy-2-naphthoate octaprenyltransferase	Metabolism; Metabolism of cofactors and vitamins; Ubiquinone and other terpenoid-quinone biosynthesis	FLKGDHHP_00175	99%	1.90 × 10^−166^
**Electron acceptor**						
*cydA*	llj:LG36_0688	cytochrome d ubiquinol oxidase subunit I	Metabolism; Energy metabolism; Oxidative phosphorylation, Environmental Information Processing; Signal transduction; Two-component system	FLKGDHHP_00735	100%	3.30 × 10^−285^
*cydB*	llx:NCDO2118_0720	cytochrome d ubiquinol oxidase subunit II	Metabolism; Energy metabolism; Oxidative phosphorylation, Environmental Information Processing; Signal transduction; Two-component system	FLKGDHHP_00736	99.7%	4.90 × 10^−184^
*cydD*	llj:LG36_0690	ATP-binding cassette, subfamily C, bacterial CydD	Environmental Information Processing; Membrane transport; ABC transporters	FLKGDHHP_00738	99.5%	0.00 × 10^0^
*cydC*	llj:LG36_0691	ATP-binding cassette, subfamily C, bacterial CydC	Environmental Information Processing; Membrane transport; ABC transporters	FLKGDHHP_00737	99.5%	0.00 × 10^0^

**Table 7 biology-14-01323-t007:** The nisin activity of *Lactococcus* sp. KTH0-1S from various cultivation conditions.

Cultivation Conditions	Inhibition Zone (mm)
Anaerobic	13.50 ± 0.87
Aerobic	13.33 ± 0.58
Anaerobic + Heme	14.00 ± 0.50
Aerobic + Heme	12.67 ± 0.29
Anaerobic + Heme + MK4	12.67 ± 0.58
Aerobic + Heme + MK4	11.00 ± 0.50

## Data Availability

The genome of *Lactococcus* sp. KTH0-1S has been submitted to the BioProject in the accession number PRJNA1274712 and the BioSample accession number SAMN48999503.

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
