# Peer review of "Genome-Driven Insights into Lactococcus sp. KTH0-1S Highlights Its Biotechnological Potential as a Cell Factory"

_biology, 2025, doi:10.3390/biology14101323_

Round 1
Reviewer 1 Report
Comments and Suggestions for Authors
The article is devoted to genome analysis, but the genome itself is not available for validation by an external researcher. The strain used in the study has also not been deposited in a publicly accessible collection. Consequently, none of the information presented in the article can be verified.
In addition, I have a number of comments: 1) Table 1 is titled ‘Primers designed and used in this study.’ Are these all primers of your authorship? If not, please provide links to the original works where these primers were first designed.
2) Lines 107–105. This describes the amplification and sequencing of a fragment of the 16S rRNA gene. The GenBank database contains such a sequence (Lactococcus lactis gene for 16S ribosomal RNA, partial sequence, strain: KTH0-1S; GenBank: AB985677.1), but it has different authors.
3) What is the point of performing PCR with primers for the nisin biosynthesis cluster if you have a sequenced genomic sequence available?
4) What is the sense of performing MLST analysis when the entire taxonomy of bacteria has long been based on genome comparison? Why did you not use resources such as JSpeciesWS or similar ones to determine the exact systematic position?
5) The section on virulence factors is very controversial. All bacteria contain a gene encoding the EF-Tu elongation factor – does this mean that they all contain a virulence factor? It is strange to read this section, especially after the introduction and lines 45-47.
6) The conclusions about genome stability are not supported by anything. The presence of prophages and mobile genetic elements has not been analysed in any way.
7) Tables 3, 4, etc. contain a strange column labelled ‘Identity’. What is the degree of similarity between what and what? Some tables also contain a column labelled ‘E-value’. What is the error value of what?
8) Tables 5 and 6 do not show the coordinates of the genes found.
9) When describing the function of the genes found, information is often given without references to the original sources. For example, lines 273-280. Did the authors themselves establish these facts in their experiments? If not, references must be provided.
10) The description of gene subsystems based on RAST analysis seems uninformative. What percentage of genes was not included in the system? Take the genome of any other lactococcus and you will get similar information. You should focus on the differences – how this strain and its genome differ (are unique) from the many other lactococcus genomes described.
11) The authors have a strange understanding of the meaning of inducible gene expression. The NICE system was developed precisely for controlled and scalable expression. If you need constitutive expression, just use another system based on a different promoter.
Author Response
Comment 1: The article is devoted to genome analysis, but the genome itself is not available for validation by an external researcher. The strain used in the study has also not been deposited in a publicly accessible collection. Consequently, none of the information presented in the article can be verified.
ANS 1: In case the external researcher would like to validate the genome of this strain, the genome of Lactococcus sp. KTH0-1S has been submitted to the BioProject in the accession number PRJNA1274712 and BioSample accession number SAMN48999503.”as mentioned in the Availability of data and materials (line 535)
Comment 2: In addition, I have a number of comments: 1) Table 1 is titled ‘Primers designed and used in this study.’ Are these all primers of your authorship? If not, please provide links to the original works where these primers were first designed.
ANS 2: Thank you for the suggestion. The primer 27F and 1492R has been cited in the table 1.
Comment 3: Lines 107–105. This describes the amplification and sequencing of a fragment of the 16S rRNA gene. The GenBank database contains such a sequence (Lactococcus lactis gene for 16S ribosomal RNA, partial sequence, strain: KTH0-1S; GenBank: AB985677.1), but it has different authors.
ANS 3: The KTH0-1S strain was previously identified as Lactococcus lactis based on 16S rRNA gene sequencing and deposited in GenBank (accession no. AB985677.1), reflecting the limitations of sequencing technology available at that time. In this study, we obtained the strain directly from the original owner group (as acknowledged) and performed whole-genome sequencing. Our genomic analysis revealed low similarity to Lactococcus lactis and other reported Lactococcus species. Based on these more comprehensive data, we report and rename this strain as Lactococcus sp. KTH0-1S, which provides a more accurate classification.3) What is the point of performing PCR with primers for the nisin biosynthesis cluster if you have a sequenced genomic sequence available?
Comment 4: What is the sense of performing MLST analysis when the entire taxonomy of bacteria has long been based on genome comparison? Why did you not use resources such as JSpeciesWS or similar ones to determine the exact systematic position?
ANS 4: We agree that whole-genome-based taxonomy, particularly ANI, has become the gold standard for determining the systematic position of bacteria. In our study, we primarily used ANI analysis to assess the taxonomic placement of Lactococcus sp. KTH0-1S, which showed an ANI value of 88.57% compared to its closest reference genome (Lactococcus cremoris GCA_004354515.1), clearly below the 95-96% threshold for species delineation. We included MLST analysis as a complementary approach to explore the phylogenetic relationship of our strain in the context of known Lactococcus type strains. Nonetheless, we have applied the genome of KTH0-1S to the JSpeciesWS platform, which showed an ANIb value of 86.5%, well below the 95% species delineation threshold, thereby supporting the MLST-based observation that KTH0-1S is genetically distinct from known Lactococcus type strains. (line 409-423)
Comment 5: The section on virulence factors is very controversial. All bacteria contain a gene encoding the EF-Tu elongation factor – does this mean that they all contain a virulence factor? It is strange to read this section, especially after the introduction and lines 45-47.
ANS 5: Thank you for your critical observation. We agree that referring to core housekeeping genes such as tuf (elongation factor Tu) as virulence factors in the pathogenic sense can be misleading. However, in the context of probiotic research, several studies have reported that certain housekeeping proteins, when surface-localized, may contribute to beneficial traits, such as adhesion to intestinal epithelial cells, biofilm formation, or immune modulation. To avoid confusion, we have reworded this section in the revised manuscript to clarify that such genes are not pathogenic virulence factors, but rather surface-associated proteins potentially supporting probiotic functions (We have additionally discussed these aspects in the revised discussion section).(line 426-440)
Comment 6: The conclusions about genome stability are not supported by anything. The presence of prophages and mobile genetic elements has not been analysed in any way.
ANS 6: We thank the reviewer for this insightful comment. We have now conducted a comprehensive analysis of MGEs and prophage regions in the Lactococcus sp. KTH0-1S genome to assess genome stability. (line 161-163) (line 284-304)
Comment 7: Tables 3, 4, etc. contain a strange column labelled ‘Identity’. What is the degree of similarity between what and what? Some tables also contain a column labelled ‘E-value’. What is the error value of what?
ANS 7: Thank you for your comment. In Tables 3, 4, and related tables, the identity value column refers to the percentage of sequence similarity between the predicted homologs in Lactococcus sp. KTH0-1S and the reference protein in the corresponding database. This identity value is generated from BLAST-p based alignment. Moreover, the E-value or expectation value indicates the statistical significance of the sequence alignment, representing the number of hits one can expect to see by chance when searching a database of a particular size. Lower E-values (closer to 0) indicate more significant matches.
Comment 8: Tables 5 and 6 do not show the coordinates of the genes found.
ANS 8: Thank you for your observation. The data presented in Tables 5 were obtained from RAST annotation, which does not provide genomic coordinates for the identified genes. Therefore, the coordinates are not available in this analysis.
Comment 9: When describing the function of the genes found, information is often given without references to the original sources. For example, lines 273-280. Did the authors themselves establish these facts in their experiments? If not, references must be provided.
ANS 9: Thank you for pointing this out. The functional descriptions of the genes are based on previously published studies and established annotations. We have revised the manuscript to include appropriate references for each gene function described in that section.( line 460, 464)
Comment 10: The description of gene subsystems based on RAST analysis seems uninformative. What percentage of genes was not included in the system? Take the genome of any other lactococcus and you will get similar information. You should focus on the differences – how this strain and its genome differ (are unique) from the many other lactococcus genomes described. (line 448-462)
ANS 10: Thank you for this valuable comment. We agree that the general description of RAST subsystems alone is not sufficiently informative. In response, we have revised the section to include the percentage of genes that were not assigned to RAST subsystem. (line 246) Moreover, we now compare the subsystem to the closest strain, L. cremoris (GCA_004354515.1) in the discussion part. (line 409-423)
Comment 11: The authors have a strange understanding of the meaning of inducible gene expression. The NICE system was developed precisely for controlled and scalable expression. If you need constitutive expression, just use another system based on a different promoter.
ANS 11: Another expression system could indeed be an option for constitutive expression. However, for food-grade recombinant protein production and as a therapeutic protein delivery vehicle, the NICE system remains the best choice, with many successful cases already developed. Therefore, the development of alternative expression hosts compatible with the NICE system can be highly valuable for these applications.
Reviewer 2 Report
Comments and Suggestions for Authors
The study identifies and characterizes a novel Lactococcus strain (KTH0-1S) with strong potential as both a probiotic and microbial cell factory.
The sequencing method section (lines 117–121) is confusing: PacBio and ONT protocols are mentioned together. Please clarify if both platforms were used, or if this is a misstatement.
For MLST and phylogenetics, specify the exact genes used for alignment and how bootstrapping or statistical support was assessed.
You list virulence-associated genes (e.g., tufA, htpB), but the Discussion later states “low number of virulence-associated genes with no significant pathogenicity risk.” This needs careful clarification. Are these genes general housekeeping/adhesion genes that overlap with VFDB hits? Otherwise, reviewers may challenge the claim of safety.
The discussion should focus on Comparative advantages of KTH0-1S vs. model strains (e.g., L. lactis NZ9000) and limitations on your study
Author Response
The study identifies and characterizes a novel Lactococcus strain (KTH0-1S) with strong potential as both a probiotic and microbial cell factory.
Comment 1: The sequencing method section (lines 117–121) is confusing: PacBio and ONT protocols are mentioned together. Please clarify if both platforms were used, or if this is a misstatement.
ANS 1: Thank you for your comment. We confirm that both PacBio and ONT platforms were used for sequencing KTH0-1S. The long-read data generated from both platforms were combined to improve genome assembly quality and completeness. We have revised the sequencing method section in the manuscript to clearly state the use of both technologies to avoid confusion and ensure clarity. (line 132-138)
Comment 2: For MLST and phylogenetics, specify the exact genes used for alignment and how bootstrapping or statistical support was assessed.
ANS 2: Thank you for your comment. We used the autoMLST for the MLST and phylogenetic analysis, which automatically selects a set of conserved housekeeping genes based on the input genome. The specific genes used for alignment are selected internally by the tool to maximize resolution and are based on core gene alignments shared among the included genomes. Phylogenetic reconstruction was carried out using the maximum likelihood (ML) method with 1,000 bootstrap replicates to assess statistical support, as per the AutoMLST default pipeline. (line 152)
Comment 3: You list virulence-associated genes (e.g., tufA, htpB), but the Discussion later states “low number of virulence-associated genes with no significant pathogenicity risk.” This needs careful clarification. Are these genes general housekeeping/adhesion genes that overlap with VFDB hits? Otherwise, reviewers may challenge the claim of safety.
ANS 3: Thank you for your comment. The identified virulence-associated genes are primarily housekeeping or adhesion-related genes that commonly appear in non-pathogenic lactic acid bacteria and are also annotated in the VFDB due to functional overlap. These genes do not contribute to pathogenicity and are essential for basic cellular functions or probiotic traits. Therefore, the presence of a low number of such genes supports the conclusion that Lactococcus sp. KTH0-1S poses no significant pathogenic risk. However, we have additionally clarified in lines 426-440.
Comment 4: The discussion should focus on Comparative advantages of KTH0-1S vs. model strains (e.g., L. lactis NZ9000) and limitations on your study
ANS 4: The comparative between KTH0-1S vs. model strains NZ9000 has been discussed in line 483-496.
Reviewer 3 Report
Comments and Suggestions for Authors
This manuscript reports the isolation and genome analysis of Lactococcus sp. KTH0-1S from Thai fermented shrimp, highlights a chromosomal nisin Z cluster (with implied auto-induction/nisin immunity), predicts respiration capability (heme-activated), and inventories CAZyme families—positioning the strain as a safe, food-grade chassis and potential NICE alternative. I have only a few minor comments.
The manuscript alternates between KTH0-1S, KTH-01S, and KYH-01S in Results—please unify to a single identifier across text/figures/supplement.
In the respiration section the text references Table 5 for cyd genes but then presents Table 6.
The identification of nisZ/lanB-C/lanK-R/nisI/ABC components appropriately suggests potential for production, regulation, and immunity. However, statements implying auto-induction and superior process economics would be stronger if supported by functional validation.
Author Response
This manuscript reports the isolation and genome analysis of Lactococcus sp. KTH0-1S from Thai fermented shrimp, highlights a chromosomal nisin Z cluster (with implied auto-induction/nisin immunity), predicts respiration capability (heme-activated), and inventories CAZyme families—positioning the strain as a safe, food-grade chassis and potential NICE alternative. I have only a few minor comments.
Comment 1: The manuscript alternates between KTH0-1S, KTH-01S, and KYH-01S in Results—please unify to a single identifier across text/figures/supplement.
ANS 1: Thank you for the suggestions. The strain label in Fig1 and 2 has been carefully corrected.
Comment 2: In the respiration section the text references Table 5 for cyd genes but then presents Table 6.
ANS : The table number has been corrected as suggestion. Line 363
Comment 3: The identification of nisZ/lanB-C/lanK-R/nisI/ABC components appropriately suggests potential for production, regulation, and immunity. However, statements implying auto-induction and superior process economics would be stronger if supported by functional validation.
ANS 3: We thank the reviewer for this valuable suggestion. We agree that the genomic identification of the nis gene cluster components indicates the potential for nisin production, regulation, and immunity. However, we acknowledge that claims regarding auto-induction and process economics should ideally be supported by experimental validation. So, in this manuscript, we only imply the potential for auto-induction benefits of this strain. Experimental validation of auto-induction is currently underway and will be reported in a subsequent publication.
Round 2
Reviewer 1 Report
Comments and Suggestions for Authors
The article has improved somewhat, but it still raises a number of questions.
You did not respond to my previous comment: 3) What is the point of performing PCR with primers for the nisin biosynthesis cluster if you have a sequenced genomic sequence available?
In addition, please note the following points:
1) The NisZ-F and NisZ-R primers are being searched for. Are you sure you developed them? Look, they appear in earlier articles by other authors.
2) I still don't see the point of MLST when the entire genome is available. Comparison of average nucleotide identity clearly indicates the taxonomic position determined to the species level. See these analysis results with your data: https://dfast.ddbj.nig.ac.jp/dqc/12afe6d4-db08-4aef-b5f3-5b939c3020d5 and 162B1076A7D0F2E87EA4 for https://jspecies.ribohost.com/jspeciesws/#analyse
3) In all tables containing genes, you must provide their coordinates or write their locus tags. When describing similarity and providing identity values, it is essential to indicate the gene/protein being compared (with its ID in the databases).
4) If you write in the text that you have found/annotated a certain gene, do not forget to indicate its locus tag as well — this will make it clearer to outside researchers which protein-coding sequence you are referring to.
Author Response
Point by point answer to Reviewer’s comment Round 2
The article has improved somewhat, but it still raises a number of questions.
Comment 1: You did not respond to my previous comment: 3) What is the point of performing PCR with primers for the nisin biosynthesis cluster if you have a sequenced genomic sequence available?
Respond 1: We thank the reviewer for the valuable comment. Although the genome of Lactococcus sp. KTH0-1S has been sequenced, PCR amplification was performed to experimentally validate the presence of key nisin biosynthetic genes. These genes are not only essential for the biosynthesis cluster but are also highly important for the functionality of the NICE expression system and the potential for auto-induction. Confirming their integrity by PCR ensures that no sequencing or assembly artifacts are present and provides assurance that these genes can be reliably amplified for future applications, such as the construction of new expression plasmids optimized for Lactococcus sp. KTH0-1S. This point has been clarified on the material and method (line 171-176)
In addition, please note the following points:
Comment 2: The NisZ-F and NisZ-R primers are being searched for. Are you sure you developed them? Look, they appear in earlier articles by other authors.
Respond 2: Thank you for the comment. For this study, we designed primers for nisZ, nisK, nisR, and nisI based on the whole-genome sequence of Lactococcus sp. KTH0-1S. However the primer’s name could be the same with previous reported.
Comment 2: I still don't see the point of MLST when the entire genome is available. Comparison of average nucleotide identity clearly indicates the taxonomic position determined to the species level. See these analysis results with your data: https://dfast.ddbj.nig.ac.jp/dqc/12afe6d4-db08-4aef-b5f3-5b939c3020d5 and 162B1076A7D0F2E87EA4 for https://jspecies.ribohost.com/jspeciesws/#analyse
Respond 2: We appreciate the reviewer’s comment. While ANI analysis indeed provides high-resolution species-level identification, we included MLST phylogenetic analysis to visualize the relationship of our strain relative to other Lactococcus strains using widely conserved housekeeping genes. MLST enables strain-level comparison across public databases, especially when ANI coverage is limited by available reference genomes. In our case, MLST reinforced that Lactococcus sp. KTH0-1S forms a separate clade, genetically distant from type strains, complementing the ANI-based identification, and we have re-clarified in the result section of Multilocus Sequence Typing (MLST) phylogenetic tree. (line 227-232), Figure 2
Comment 3: In all tables containing genes, you must provide their coordinates or write their locus tags. When describing similarity and providing identity values, it is essential to indicate the gene/protein being compared (with its ID in the databases).
Respond 3: Thank you for your valuable comment. We have revised all relevant tables to include the genomic coordinates for each listed gene, specifying the reference gene/protein used for comparison, along with its corresponding database accession number (Table 3, 4, 6 and S1). However, Table 5 presents the EPS gene cluster identified in the genome based on RAST annotation, which does not provide corresponding locus tags or similarity/identity values for each gene.
Comment 4: If you write in the text that you have found/annotated a certain gene, do not forget to indicate its locus tag as well — this will make it clearer to outside researchers which protein-coding sequence you are referring to.
Respond 4: Thank you for the comment. We agree that including locus tags improves clarity. As suggested, we have added the corresponding locus tags or ORF names based on annotation tools to all relevant tables (Table 3, 4, 6 and S1). This ensures that when specific genes are mentioned in the text or tables, readers can directly trace them back to the exact annotated coding sequence in the genome.